# Regulation of *tert*-Butyl Hydroperoxide Resistance by Chromosomal OhrR in *A. baumannii* ATCC 19606

**DOI:** 10.3390/microorganisms9030629

**Published:** 2021-03-18

**Authors:** Shih-Jie Chen, Hung-Yu Shu, Guang-Huey Lin

**Affiliations:** 1Master Program in Microbiology and Immunology, School of Medicine, Tzu Chi University, Hualien 97004, Taiwan; 105329103@gms.tcu.edu.tw; 2Department of Bioscience Technology, Chang Jung Christian University, Tainan 71101, Taiwan; hyshu@mail.cjcu.edu.tw; 3International College, Tzu Chi University, Hualien 97004, Taiwan

**Keywords:** *Acinetobacter baumannii*, Ohr, OhrR, organic hydroperoxide, antibiotic resistance, bacterial viability, bacterial virulence

## Abstract

In this study, we show that *Acinetobacter baumannii* ATCC 19606 harbors two sets of *ohrR-ohr* genes, respectively encoded in chromosomal DNA and a pMAC plasmid. We found no significant difference in organic hydroperoxide (OHP) resistance between strains with or without pMAC. However, a disk diffusion assay conducted by exposing wild-type, *∆ohrR-C*, *C* represented gene on chromosome, or *∆ohr-C* single mutants, or *∆ohrR*-*C∆ohr*-*C* double mutants to *tert*-butyl hydroperoxide (*t*BHP) found that the *ohrR-p-ohr-p* genes, *p* represented genes on pMAC plasmid, may be able to complement the function of their chromosomal counterparts. Interestingly, *∆ohr-C* single mutants generated in *A. baumannii* ATCC 17978, which does not harbor pMAC, demonstrated delayed exponential growth and loss of viability following exposure to 135 μg of *t*BHP. In a survival assay conducted with *Galleria mellonella* larvae, these mutants demonstrated almost complete loss of virulence. Via an electrophoretic mobility shift assay (EMSA), we found that OhrR-C was able to bind to the promoter regions of both chromosomal and pMAC *ohr*-*p* genes, but with varying affinity. A gain-of-function assay conducted in *Escherichia coli* showed that OhrR-C was not only capable of suppressing transformed *ohr-C* genes but may also repress endogenous enzymes. Taken together, our findings suggest that chromosomal *ohrR-C-ohr-C* genes act as the major system in protecting *A. baumannii* ATCC 19606 from OHP stresses, but the *ohrR-p-ohr-p* genes on pMAC can provide a supplementary protective effect, and the interaction between these genes may affect other aspects of bacterial viability, such as growth and virulence.

## 1. Introduction

Bacteria are often exposed to reactive oxygen species (ROS) and other organic hydroperoxides (OHPs) produced by host phagocytic cells [1] as part of the immune response [2,3]. OHPs produced by free radical-catalyzed oxidation of polyunsaturated fatty acids (PUFAs) from the host cell [2] are highly toxic molecules that can generate organic free radicals and induce oxidative injury to bacterial cell components. Bacteria have therefore evolved several strategies to protect themselves against oxidative stress. One such strategy is to produce enzymes that can directly detoxify OHPs by transforming them into unreactive alcohols. Ohr (organic hydroperoxide resistance protein) is a thiol peroxidase that is central to the bacterial response against OHPs [4,5], and it acts to neutralize OHPs via a redox-active disulfide bond formed by two conserved cysteines at the catalytic site [6]. Two homologs of Ohr, respectively termed OhrA and OhrB, have been identified in *Bacillus subtilis* [7].

The expression of Ohr enzymes is suppressed by OhrR (organic hydroperoxide resistance regulator) [8]. OhrR belongs to the MarR superfamily of transcription factors and is found in both Gram-negative and Gram-positive bacteria [9]. The oxidation of a highly-conserved cysteine in the N-terminus of OhrR upon exposure to OHPs disrupts its DNA binding activity [10], leading to the derepression of *ohr* gene. In *B. subtilis*, the oxidation of the sole cysteine residue in OhrR allows the regulator to retain its DNA-binding activity, and the derivative from an additional oxidation is required for derepression [11,12]. In *Burkholderia thailandensis*, OhrR oxidation results in the formation of a reversible disulfide bond between conserved cysteines in the N-terminus and C-terminus of separate monomers resulting for attenuation of DNA binding in vitro [13].

*Acinetobacter baumannii*, a type of gammaproteobacteria of the Moraxellaceae family, is widely found in soil, water, animals, and food items, and has gained increasing attention in recent years due to its role in nosocomial infections [14,15]. *A. baumannii* can induce a vast range of serious infections, including pneumonia, bloodstream infections, urinary tract infections, wound and burn infections, and secondary meningitis [15,16,17,18]. According to genomic data from the National Center for Biotechnology Information (NCBI), as of January 2021, genome assemblies for at least 4914 strains of *A. baumannii* have been deposited, and high genomic diversity between strains has been observed. The most studied *A. baumannii* strain, ATCC 17978, was isolated from a four-month-old infant with fetal meningitis, while the ATCC 19606 strain used in this study was isolated from urine [19]. Plasmid availability is one of the most important distinctions between these two strains, as *A. baumannii* ATCC 19606 harbors pMAC, a 9540-base pair (bp) plasmid that contains 11 annotated open reading frames (ORFs) [20]. ORF8 and ORF9 have been respectively annotated as *ohrA* and *ohrR*, and this has been confirmed through functional analysis in *Escherichia coli* [20].

In this study, we identified the presence of another *ohrR*-*C*-*ohr*-*C* gene cluster in the chromosomal DNA of *A. baumannii* ATCC 19606 through in silico analysis, and subsequently investigated the function and significance of this newly-discovered set of *ohr*-*c* genes. Our results show that chromosomal *ohrR-C-ohr-C* may play a greater role in resistance to OHPs, and this may have implications for the study of *ohr* genes in other bacterial species, as well as the development of novel antibacterial therapies against *A. baumannii* infections.

## 2. Material and Methods

### 2.1. Bacterial Strains and Culturing Conditions

*A. baumannii* ATCC 19606 and *E. coli* strains were grown in LB medium [21] at 37 °C with shaking. All of the strains were listed in Table 1. LB medium with 1.5% agar was prepared for solid cultures. Antibiotics were used at different concentrations according to the cultured strains.

The growth curve of each strain was determined according to culture turbidity, using optical density measurements at 600 nm (OD_600_). Viable bacteria counts for each strain were determined using the drop plate method, as previously described [22]. At each time point, 5 μL of 10-fold serially diluted bacteria was dropped on LB agar plates containing ampicillin to establish viable bacteria counts. Bacterial growth curves were determined every three hours for the first 12 h, after which bacterial viability was assessed every 12 h, until 48 h had elapsed from the initial drop.

**Table 1 microorganisms-09-00629-t001:** Plasmids and bacterial strains used in this study.

Plasmid	Description	Antibiotic Resistance (µg/mL)	Reference/Source
pK18mobsecB	Suicide vector for homologous recombination	Km50	[23]
pK18dohrR	pK18mobsecB contains the upstream and downstream regions of *ohrR-C*	Km50	This study
pK18dohr	pK18mobsecB contains the upstream and downstream regions of *ohr-C*	Km50	This study
pK18dohrRohr	pK18mobsecB contains the upstream and downstream regions of *ohrR-C-ohr-C*	Km50	This study
pQE80L	Expression vector with *colE1* origin for His-tag fusion protein purification	Amp50	Qiagen
pOhrR-C	*ohrR-C* gene with N-terminal-fused His-tag in pQE80L	Amp50	This study
pOhrR-C-P_ohr_-ohr-C	*ohr-C* and *ohrR-C* gene promoter was cloned downstream of pOhrR-C	Amp50	This study
**Strain**	**Description**		**Reference/Source**
*E. coli* DH5α	F^−^, *supE44*, *hsdR17*, *recA1*, *gyrA96*, *endA1*, *thi-1*, *relA1*, *deoR*, λ^−^		ATCC 53868
*E. coli* S17-1λ*pir*	*thi*-, *thr* , *leu* , *tonA* , *lacY* , *supE* , *recA::RP4-2-Tc::Mu, Smr, lpir*		[23]
*Acinetobacter baumannii* ATCC 19606	Primary strain used in this study		[19]
*Acinetobacter baumannii* ATCC 17978	Most studied strain to date		[14]
*∆ohrR*	Marker-less *ohrR-c* deletion mutant		This study
*∆ohr*	Marker-less *ohr-c* deletion mutant		This study
*∆ohrRohr*	Marker-less *ohrR-c-ohr-c* deletion mutant		This study

Amp: ampicillin; Km: kanamycin.

### 2.2. Markerless Mutant Generation

Both *ohr-C* and *ohrR-C* mutants were generated by markerless gene deletion. The upstream and downstream DNA fragments of the respective genes were amplified by PCR, using the primers listed in Table 2, and were cloned into the *Bam*HI and *Hind*III sites of plasmid pK18mobsecB (Table 1) [23] using the Gibson assembly cloning kit (New England Biolabs; Ipswich, MA, USA). The resulting plasmid was transformed into *E. coli* S17-1λ*pir* to produce a donor for conjugation with *A. baumannii*. This plasmid DNA was maintained in the chromosomal region near *ohr-C* through the first homologous recombination event, and it was selected using the kanamycin resistance gene. Excision of the resulting plasmid DNA by the second crossover event was facilitated by selection on medium containing 10–20% sucrose but without kanamycin. The deletion mutant was obtained without any markers and was subsequently confirmed by PCR analysis.

### 2.3. Minimum Inhibition Concentration

The resistance of each strain to antibiotics was assessed by liquid minimum inhibition assay. Specifically, each antibiotic was inoculated into 96-well microtiter plates with 2-fold serial dilution. Each strain was cultured in 3 mL of LB medium at 37 °C overnight. Diluted overnight cultures with an OD_600_ of 0.1 were then inoculated into 96-well microtiter plates. Optical density was determined after overnight culture. The MIC was defined as the lowest concentration of antibiotics that inhibit bacterial proliferation [24].

### 2.4. Disk Diffusion Assay

The resistance of each strain to organic hydroperoxide was evaluated by disk diffusion assy. In brief, each strain was cultured in 3 mL of LB medium at 37 °C overnight. Diluted overnight cultures with an OD_600_ of 0.1 were then densely inoculated onto LB agar plates using a cotton swab, in order to ensure the confluent growth of each strain. Whatman filter paper discs imbued with 5 μL of 300 μM *t*BHP were aseptically applied to the surface of the agar plate, and the zones of growth inhibition were measured after incubation for 16 h at 37 °C.

### 2.5. RNA Extraction and qRT-PCR

Each strain was cultured in LB medium with agitation at 37 °C overnight. Bacteria were sub-cultured in a fresh 50 mL medium for three hours. Samples were collected for the non-treatment group and mixed with 0.1 volume of fixing solution (5% acid phenol, 95% ethanol). After centrifugation by 17,000× *g* at 4 °C, the supernatant was discarded, and the remaining cell pellets were stored at −80 °C for RNA extraction.

Cell pellets were thawed on ice and resuspended in 1 mL of NucleoZOL (MACHEREY-NAGEL; Düren, Germany), mixed thoroughly with 400 μL of diethyl pyrocarbonate (DEPC)-treated H_2_O, and then incubated at room temperature for 15 min. The supernatant was recovered after centrifugation at 17,000× *g* at 4 °C for 20 min, then fully mixed with 5 μL of 100% 4-bromoanisole and incubated at room temperature for 10 min. Excess protein was removed by centrifugation at 17,000× *g* at 4 °C for 20 min. The RNA suspension was subsequently mixed with an equal volume of isopropanol for 15 min for RNA precipitation. The RNA pellet was washed twice with ice-cold 75% ethanol and resuspended in 30 μL of DEPC-treated H_2_O for the following analysis.

A total of 2 μg of RNA was subsequently used to prepare cDNA after a Nanodrop spectrophotometer (NanoDrop 2000C, Thermo Fisher Scientific; Waltham, MA, USA) analysis was conducted to determine RNA concentrations. The qRT-PCR mixture contained 10× reaction buffer, 200 U of MMLV high performance reverse transcriptase (Epicentre; Madison, WI, USA), 100 mM of dithiothreitol (DTT), 2.5 mM dNTP, and 1 nM of hexamer. The reaction was conducted in a LightCycler^®^ 480 (Roche; Basel, Switzerland). Gene-specific primers used to determine the presence and expression levels of the *ohr-C*, *ohrR- C*, *ohr-p*, and *ohrR-p* genes are listed in Table 2. The gyrase gene served as an internal control, and was amplified by PCR using the specific primers, *gyr*F and *gyr*R (Table 2) [25].

### 2.6. OhrR Overexpression and Purification

The PCR product of the *ohrR-C* gene was amplified using ohrR-c-F and ohrR-c-R primers, and cloned into the *Bam*HI and *Kpn*I sites of plasmid pQE80L (Qiagen, Hilden, Germany) to generate pOhrR-C. To express OhrR-C protein, *E. coli* DH5α(pOhrR-C) was expanded at 37 °C by inoculating a 0.5-mL overnight culture into 50 mL of LB medium containing ampicillin. Incubation was continued at 37 °C until the culture reached an OD_600_ of 0.6. Protein expression was induced by adding IPTG to achieve a final concentration of 1 mM. After incubation overnight at 37 °C, the cells were harvested by centrifugation at 4000× *g* for 15 min. The cells were then stored at −80 °C until use.

OhrR-C protein purification was performed by a method described elsewhere [26]. Frozen cells overexpressing OhrR-C were suspended in 30 mL of 1× binding buffer containing 5 mM imidazole, 0.5 mM NaCl, and 20 mM Tris-HCl (pH 8.0) and subjected to 3 cycles of freezing and thawing at –80 °C and room temperature, respectively. The thawed cells were disrupted by high pressure at 4 °C using a low-temperature cell disruptor, JNBIO JN-O2C (Guangzhou, China). The cell extract was separated from the cell debris by centrifugation at 17,000× *g* for 30 min at 4 °C (Avanti J-25 Centrifuge, JA25.5 rotor, Beckman Coulter; Brea, CA, USA). The OhrR-C containing cell extract was then purified by Ni-affinity chromatography (Novagen; Madison, WI, USA). Purified fractions were analyzed via 15% sodium dodecyl sulfate polyacrylamide gel electrophoresis (SDS-PAGE) and stained with Coomassie Brilliant Blue G-250.

### 2.7. EMSA

The DNA fragment of the *ohr-C* promoter and *ohr-p* promoter were amplified by PCR, using gene-specific primers (Table 2). EMSA was performed as previously described, with some modifications [26]. The reaction mixtures for the binding assays contained different concentrations of OhrR-C protein. The binding reactions were performed in binding reaction buffer (20 mM Tris-HCl (pH 7.5), 100 mM MgCl_2_, 150 mM KCl, 50 mM EDTA, 12.5% glycerol) supplemented with 250 μM DTT, 830 ng/mL poly(dI-dC) and 250 ng/mL bovine serum albumin (BSA). The reaction mixtures were incubated for 30 min at room temperature before adding 50 nM of DNA fragments. Samples were incubated for another 30 min, they were mixed with an equal volume of sample buffer (62 mM Tris-HCl, 0.01% bromophenol blue, and 10% glycerol) and loaded onto 8% non-denaturing polyacrylamide gels containing 0.5× Tris-borate-EDTA buffer. Electrophoresis was performed at 100 V for 1–1.5 h at 4 °C. Images were captured after the gels were stained with SYBR Gold Nucleic Acid Gel Stain (Thermo Fisher Scientific; Waltham, MA, USA) for 20 min at room temperature, and imaged using the Ultra Slim LED Illuminator (MAESTROGEN; Hsinchu City, Taiwan). A DNA fragment of the gyrase gene promoter from *A. baumannii* was amplified by PCR, using the specific primers *gyr-p*F and *gyr-p*R (Table 2), and was used as a negative (non-specific) control.

### 2.8. G. Mellonella Experiments

A virulence comparison was performed for *A. baumannii* ATCC 19606, ATCC 17978 and the *ohr-C* or *ohrR-C* mutants of each strain. All the procedures were performed as previously described, with minor modifications [27]. Overnight cultures of each strain were washed twice with PBS (0.137 M NaCl, 2.7 mM KCl, 10 mM Na_2_HPO_4_, 1.8 mM KH_2_PO_4_), then diluted in PBS. Ten *G. mellonella* larvae were selected for the same total weight and were kept in Petri dishes without food prior to infection. Each larva was infected with 5 × 10^6^ colony-forming units (CFU) of each strain. Bacteria in 10-mL aliquots were injected into the hemocoel of each larva via the last left proleg by a Hamilton syringe. Infected larvae were incubated at 37 °C and scored for survival (alive/dead) every 24 h. Larvae were also scored for melanization over 96 h, according to a previously described scoring method [28].

## 3. Results

### 3.1. Identification of Two Paralogous ohr-ohrR Genes in A. baumannii ATCC 19606

Previous research showed that the pMAC plasmid in *A. baumannii* ATCC 19606 contains 11 ORFs, with ORF8 and ORF9 respectively annotated as putative OhrA and OhrR enzymes that can contribute to *tert*-butyl hydroperoxide (*t*BHP) resistance [20]. Using an in silico analysis, we proceeded to identify another set of putative Ohr-C (DJ41_1043) and OhrR-C (DJ41_1042) proteins on the chromosomal DNA of *A. baumannii* ATCC 19606, and these shared 44.4% and 41.3% identity with ORF8 and ORF9 on pMAC. Interestingly, the chromosomal *ohrR-C* gene is located upstream of the chromosomal *ohr-C* gene, and the genetic architecture suggests that each gene possesses its own promoter regions, and can be transcribed independently in the same direction (Figure 1); however, on pMAC, ORF8 is located upstream from ORF9, and considering that the start codon of ORF9 is located only 7 bp downstream of the stop codon of ORF8, it is possible that both genes may be transcribed on the same mRNA (Figure 1).

We proceeded to label ORF8 as Ohr-p and ORF9 as OhrR-p, in order to distinguish them from their chromosomal counterparts. We found that Ohr-p shared 44.4% amino acid similarity with chromosomal Ohr (Ohr-C) and 41.5% similarity with OhrA of *Xanthomonas campestris* pv. *phaseoli*. Ohr-C was found to share 37.1% similarity with Ohr of *E. coli* (Ohr-Ec), and 56.3% similarity with OhrA of *X. campestris* (Ohr-Xc) (Appendix A). Chromosomal OhrR (OhrR-C) shared 41.3% amino acid similarity with OhrR-p, and 51.5%, 48.6%, and 45.9% similarity with OhrR of *X. campestris* (OhrR-Xc), *Pseudomonas aeruginosa* (OhrR-Pa), and *B. subtilis* (OhrR-Bs), respectively (Appendix A). Secondary structure analysis of OhrR-C revealed six α-helices and two β-sheets, consistent with the typical structure of a MarR protein (Appendix A).

### 3.2. Role of ohr-p and ohrR-p in tBHP Resistance Is Limited

Several methods have been applied by our lab (data not shown) and others [20] to evaluate the role of pMAC in organic peroxide resistance by plasmid curing, but these efforts were not successful since we can find pMAC in bacteria after curing process. To screen for the presence of chromosomal *ohrR-C-ohr-C* and *ohr-p-ohrR-p* genes on pMAC in different *Acinetobacter* strains for which genomic sequences are not available but identified by 16S rRNA gene amplification and sequences analysis. Gene-specific primers were designed for gene amplification (Table 2). Subsequent PCR results revealed that most *Acinetobacter* spp. had chromosomal *ohrR-C-ohr-C* genes, except *A. soli* (Figure 2A, lane 7). However, *ohr-p-ohrR-p* genes on pMAC were only found in *A. baumannii* ATCC 19606 and *A. baumannii* ATCC 15308 (Figure 2A, lanes 8, 9, 10). Growth conditions in the presence of 135 μg of *t*BHP were assessed for each of these strains, using the disk diffusion assay, but no significant difference in *t*BHP resistance was observed between strains with or without the pMAC plasmid (Figure 2B). Different concentrations of *t*BHP and cumene hydroperoxide were also tested, but the results were similar to those observed with 135 μg of *t*BHP (data not shown). This suggests that the *ohr-p-ohrR-p* genes on pMAC have a limited role in *t*BHP resistance, and we therefore focused on elucidating the functional role of chromosomal *ohrR-C-ohr-C* genes in subsequent experiments.

### 3.3. Chromosomal ohr-C Plays an Important Role in Bacterial Proliferation

To understand the importance of chromosomal *ohrR-C-ohr-C* genes for *A. baumannii* ATCC 19606, we constructed single and double *ohrR-C* and *ohr-C* mutants by markerless allelic exchange. Bacteria used in these experiments were all cultured in lysogeny broth (LB), and the optical density and viable count of cultures was determined every three hours over a 24-h period, in order to obtain growth curves for each strain. Results revealed no significant difference in growth curves for wild-type and mutant *A. baumannii* ATCC 19606 (Figure 3A). In addition, *ohrR-C* and *ohr-C* single mutants were also generated in *A. baumannii* ATCC 17978, and growth curves showed that the *ohr-C* mutant underwent a delay in entering the exponential growth phase during the first 12 h, but eventually reached comparable levels of growth with wild-type after entering the stationary phase (Figure 3B). These results suggest an important role for chromosomal *ohr-C* in overcoming organic peroxide stress during the exponential growth phase for *A. baumannii* ATCC 17978. However, we did not observe similar results with *ohr-C* single and *ohrR-C-ohr-C* double mutants of *A. baumannii* ATCC 19606 cultured in LB medium for 24 h, implying that the *ohr-p-ohrR-p* genes on pMAC may act to reduce organic peroxide stress (Figure 3A).

Antibiotic testing was used to determine the minimum inhibition concentration of different strains, and the results showed that there were no significant differences between *A. baumannii* ATCC 19606 strains. However, loss of *ohrR-C* increased kanamycin resistance in *A. baumannii* ATCC 17978, while the *ohr-C* mutant became more susceptible to antibiotics tested (Figure 3C). This suggests that the *ohr-p-ohrR-p* genes on pMAC may enhance resistance by mitigating organic peroxide stress generated during antibiotic treatment.

### 3.4. Gene Expression of Chromosomal ohr-C Can Be Induced by tBHP

To assess gene expression levels for chromosomal *ohrR-C*-*ohr-C* and pMAC *ohr-p*-*ohrR-p* genes, quantitative reverse transcription PCR (qRT-PCR) was conducted to analyze gene expression in bacteria with or without *t*BHP treatment. *A. baumannii* wild type, *ohrR-C*, *ohr-C* single and double mutants were sub-cultured in LB medium at an initial OD_600_ of 0.05 for 3 h, and then treated with 200 μM of *t*BHP for 20 min prior to acid phenol fixation. Samples were stored at −80 °C if RNA extraction was not performed immediately. Gene-specific primers (Table 2) were used to perform qRT-PCR, and the results showed that in the presence of *t*BHP, *ohr-C* and *ohr-p* expression respectively increased by 1.76-fold and 4.12-fold over untreated controls (Figure 4A). In *∆ohrR-C* mutants, *ohr-C* expression increased more than 100-fold over wild-type. Both *ohrR-p* and *ohr-p* did not demonstrate significantly elevated gene expression in all mutants (Figure 4B). In *∆ohrR-C* mutants, *t*BHP induced 50-fold higher chromosomal *ohr-C* expression over wild-type strains, indicating that OhrR-C has a repressive effect on *ohr-C* expression, but little effect on *ohr-p* expression (Figure 4C). As for the expression of *ohr-p-ohrR-p* genes on pMAC in the presence of *t*BHP, neither *ohr-p* nor *ohrR-p* gene expression was strongly induced in *ohr-C* single mutants or *ohrR-C-ohr-C* double mutants (Figure 4C). These results indicate that even with the loss of *ohrR-C-ohr-C*, the *ohr-p* genes on pMAC will not be upregulated to overcome the effects of *t*BHP treatment. To further clarify the role of chromosomal *ohrR-C-ohr-C*, mutants were generated in a strain without pMAC (*A. baumannii* ATCC 17978), and qRT-PCR was conducted to assess gene expression with or without *t*BHP treatment. The results were similar to those observed for *A. baumannii* ATCC 19606, in which *ohr-C* expression was induced by *t*BHP (Figure 5A) and repressed by OhrR-C (Figure 5B).

A disk diffusion assay was conducted to assess *t*BHP resistance for each strain of *A. baumannii* 19606. The zone of inhibition was diminished in *ohrR-C* mutants as compared with wild-type (Figure 6A), and this observation concurs with the qRT-PCR results. A previous study has shown that the *ohr-p*-*ohrR-p* genes on pMAC contribute to organic peroxide resistance in *A. baumannii* 19606 [20], and to better ascertain the role of the pMAC genes, a chromosomal *ohrR-C-ohr-C* double mutant was generated by markerless allelic exchange, to create strains with *ohr-p-ohrR-p* only. We found that pMAC alone did not contribute to *t*BHP resistance (Figure 6A). Moreover, a disk diffusion assay conducted with *ohrR-C* and *ohr-C* single mutants of *A. baumannii* ATCC 17978 found that the *ohr-C* single mutant had reduced resistance against *t*BHP (Figure 6B). These results further confirm the importance of *ohr-C* to *t*BHP resistance.

### 3.5. Binding of OhrR-C to the Promoter of the ohr-C and ohr-p Genes

To confirm that chromosomal OhrR-C suppressed *ohr-C* gene expression, EMSA was conducted to ascertain the direct binding of Ohr-C to the promoter region of the *ohr-C* gene. The putative OhrR-C box of *A. baumannii*, AAATXAT-14-ATXTATTT (Figure 7) contains an AT-rich motif that has been found in most of the DNA-binding sequences of OhrR-C homologs [7,13,29,30,31]. The entire 85-bp putative promoter region of the *ohr-C* gene was split into different fragments (Figure 7). P_ohr-1_ and P_ohr-2_ divided this region into two parts, each covering half of the promoter region. P_ohr-3_ is 18-bp shorter than P_ohr_ (Figure 8A). We incubated 50 nM of DNA probes with recombinant OhrR-C purified from an *E. coli* overexpression strain. The results showed that only P_ohr-3_ and P_ohr_ could be bound by 600 nM of OhrR-C, with a mobility shift in electrophoresis observed (Figure 8A). We then used different concentrations of OhrR-C to interact with P_ohr_. Results revealed that a mobility shift could be observed with just 400 nM of OhrR-C mixed with DNA (Figure 8B, lane 4). Increasing OhrR-C concentration to 600 nM led to a significant mobility shift (Figure 8B, lane 6). The DNA/protein complexes became trapped in electrophoresis wells when the amount of OhrR-C exceeded 800 nM (Figure 8B, lane 8).

Recombinant chromosomal OhrR-C was also used to test binding capability with the promoter region of *ohr-p* on pMAC. A mobility shift was observed when 1 μM of OhrR-C was mixed with DNA (Figure 8C, lane 4). Increasing OhrR-C concentration to 1.4 μM led to a significant mobility shift (Figure 8C, lane 5), and the DNA/protein complexes became trapped in the well when the amount of OhrR-C exceeded 2.6 μM (Figure 8C, lane 8). Taken together, recombinant chromosomal OhrR-C showed less affinity for the promoter region of *ohr-p* on pMAC, compared with chromosomal *ohr-C*.

### 3.6. OhrR-C Suppressed Ohr-C Expression in E. coli

We sought to examine the role of *ohrR-C-ohr-C* genes by cloning them into a pQE80L plasmid, and then transformed the plasmid into *E. coli* to ascertain whether gain of *t*BHP resistance functions could be observed in the transformants. We therefore cloned *ohrR*-C into pQE80L under the isopropyl β-D-1-thiogalactopyranoside (IPTG)-inducible T5 promoter with N-terminal 6-His fusion, to generate the pQE80L_OhrR-C plasmid. Subsequently, *ohr-C* with its endogenous promoter was constructed downstream of *ohrR-C*, with a FLAG-tagged C-terminal, to generate the pQE80L_OhrR-C_Ohr-C plasmid (Figure 9A). Each plasmid was transformed into the *E. coli* DH5α strain for *t*BHP resistance tests. *E. coli* transformed with an empty pQE80L showed no difference in *t*BHP resistant properties, but in transformants with OhrR-C induced by 1 mM IPTG, increased *t*BHP sensitivity was observed, suggesting that OhrR-C may also suppress the *t*BHP resistance genes in *E. coli.* However, transformants with the *ohr-C* gene had reduced inhibition zones, indicative of increased *t*BHP resistance; but following the induction of OhrR-C expression, Ohr-C was subsequently repressed, and increased *t*BHP sensitivity was observed (Figure 9B). From this gain-of-function experiment, it was observed that OhrR-C was able to regulate the expression of the *ohr-C* gene derived from *A. baumannii*. 

### 3.7. The ohrR-C Mutant Demonstrated Decreased Virulence in Galleria mellonella

To assess the overall virulence of *ohr-C* and *ohrR-C* mutants compared with wild-type, *G. mellonella* larvae (n = 10) were respectively incubated with 5 × 10^6^ colony-forming units (CFU) of each *A. baumannii* strain. We observed that *ohrR-C* mutants displayed significantly greater virulence in *G. mellonella* as compared to wild-type, *ohr-C* single mutants, and *ohrR-C-ohr-C* double mutants, with only 10% larvae survival at 72 h compared to 70% survival for the other three strains. However, *ohr-C* single mutants and *ohrR-C-ohr-C* double mutants retained 60% virulence in *G. mellonella* at 96 h, indicating that the *ohr-p-ohrR-p* genes on pMAC may play a role in virulence for *G. mellonella* (Figure 10A). A survival assay was performed with *A. baumannii* ATCC 17978 strains without pMAC, to better understand the importance of chromosomal *ohrR-C* and *ohr-C* for virulence against *G. mellonella*. The results showed that, while wild-type strains killed all infected larvae within 24 h, the *ohr-C* single mutants had significantly reduced virulence, even at 96 h after infection (Figure 10B). The results suggest that the chromosomal *ohr-C* gene plays an important role in *A. baumannii* virulence toward *G. mellonella*, and this may have implications for other hosts as well.

## 4. Discussion

The Ohr protein is a thiol-dependent peroxidase that plays a major role in the response of bacteria against organic peroxide [29]. Most bacteria harbor at least one Ohr protein [30,31]. In this study, both chromosomal and plasmid *ohr* genes were identified in *A. baumannii* 19606. The promoter regions of both *ohr* genes can be bound by OhrR, albeit with varying affinity. Other examples of bacteria with more than one *ohr* gene have previously been identified. In *B. subtilis*, two homologs of Ohr, OhrA and OhrB, have been identified and studied [7]. OhrR is capable of repressing *ohrA* expression, while *ohrB* is unaffected [7]. However, in *P. aeruginosa*, homologous OhrR and OspR have been shown to act as regulatory switches in response to oxidative stress [32]. OspR regulates *gpx*, which encodes a glutathione peroxidase. In an *ospR* mutant, in which *gpx* is depressed, the induction of *ohr* expression by oxidative stress is reduced. Similarly, in an *ohrR* mutant, where *ohr* is derepressed, organic hydroperoxide induction of *gpx* is reduced [32].

Several thousand strains of *A. baumannii* have been sequenced, and some of these have been reported to harbor plasmids [20]. Genomic sequence analysis revealed that pM1301-3 of *Acinetobacter* sp. M131 and a plasmid in *A. baumannii* ab736 had comparable length and shared high sequence similarity with pMAC of *A. baumannii* ATCC 19606. All of these plasmids contained *ohr-p-ohrR-p* genes. Multi-sequence alignment also revealed that *ohr-p-ohrR-p* genes can be found in pALWS1.1 of *Acinetobacter lwoffii* VS15 and p1AsACE of *Acinetobacter schindleri* ACE. These observations suggest that *A. baumannii* may gain these genes through horizontal gene transfer. Based on our observation of growth curves for the *ohr-C* mutant of *A. baumannii* ATCC 17978, chromosomal *ohr-C* not only plays a role in generating resistance to OHPs but is also associated with rapid bacterial growth (Figure 3B). However, this was not observed for the *ohr-C* single mutant and *ohrR-C-ohr-C* double mutant of *A. baumannii* ATCC 19606 (Figure 3A), and this may be due to the complementary effect of the *ohr-p-ohrR-p* genes on pMAC.

Both *A. baumannii* ATCC 19606 [33] and *A. baumannii* ATCC 17978 [34] harbor resistance to multiple antibiotics, but with different resistance profiles. For example, previous research [34] and our unpublished data demonstrate that *A. baumannii* ATCC 19606 has higher minimal inhibitory concentrations (MIC) with tigecycline (2 mg/L), imipenem (2 mg/L), and colistin (1 mg/L), as compared to *A. baumannii* ATCC 17978 [34]. Some bactericidal antibiotics operate through the production of highly deleterious hydroxyl radicals in bacteria [35]. Surprisingly, lipoyl groups are essential for parasite survival in host cells [36,37] and for virulence [38]. Taken together, our findings suggest that *A. baumannii* ATCC 19606 may have gained pMAC via horizontal gene transfer, and this strain exhibits higher MIC to antibiotics not only because of the antibiotic resistance genes encoded on pMAC, but also because two sets of *ohr* genes are present in the chromosomal DNA and pMAC respectively. This can increase resistance to OHPs and hydroxyl radicals from bactericidal antibiotics, thereby increasing virulence. Future research into novel antibacterial therapies that target these *ohr* genes may therefore be warranted.

## 5. Conclusions

*A. baumannii* has gained increasing attention in recent years due to its role in nosocomial infections. The strains examined in this study all harbor resistance to multiple antibiotics, and considering that some antibiotics act by producing hydroxyl radicals that are highly deleterious, it is possible that aside from antibiotic resistance genes present on plasmids, the presence of *ohr* genes on plasmids and chromosomal DNA can contribute to bacterial viability and resistance as well. Therefore, a study of the *ohr* genes and their functions may have important implications for understanding bacterial resistance and preventing or addressing nosocomial infections.

## Figures and Tables

**Figure 1 microorganisms-09-00629-f001:**
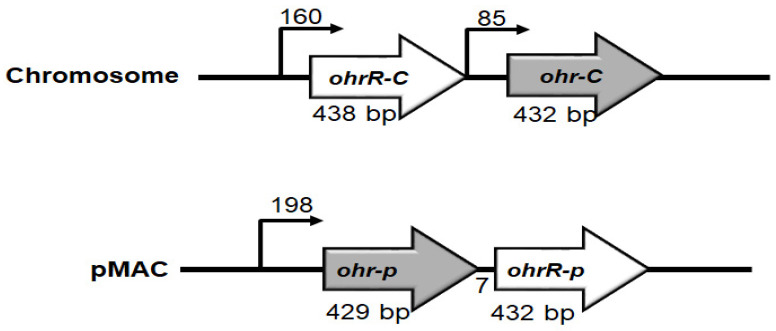
Genetic organization of *ohr-ohrR* genes on chromosomal DNA and pMAC. Numbers above bent arrows indicate the distance in base pairs between two genes.

**Figure 2 microorganisms-09-00629-f002:**
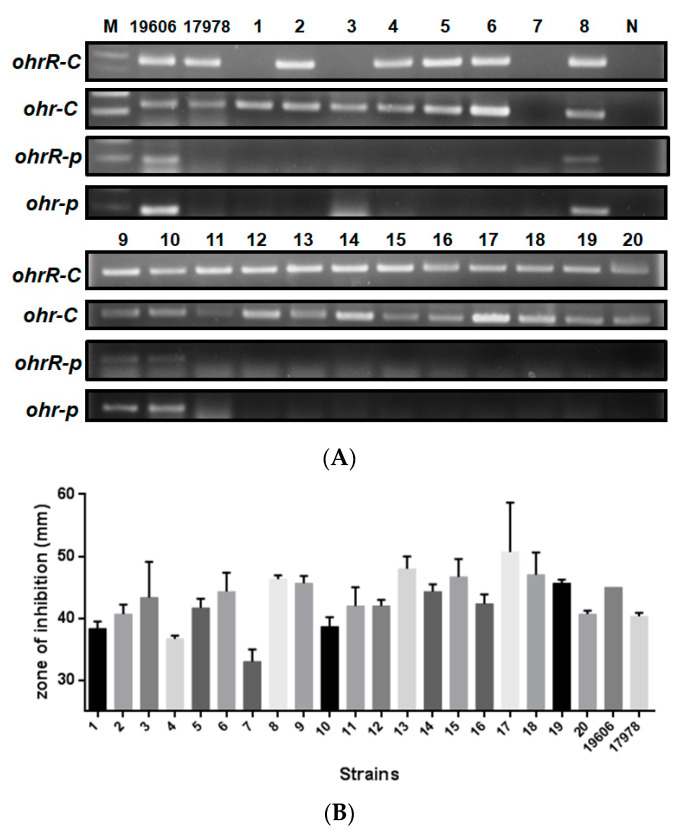
Presence of *ohrR-ohr* genes and *t*BHP resistance of 20 *Acinetobacter* isolates. (**A**) PCR products amplified by primers specific to chromosomal or pMAC *ohrR-ohr* genes. (**B**) Inhibition zones for each isolate following treatment with 135 μg of *t*BHP in a disk diffusion assay. Numbers 1–20 indicate the strain numbers of *Acinetobacter* isolates. M is a molecular weight marker. N is a negative control using ddH_2_O as a template. 19606 and 17978 indicate *A. baumannii* ATCC 19606 and ATCC 17978, respectively. These data were obtained from three independent experiments.

**Figure 3 microorganisms-09-00629-f003:**
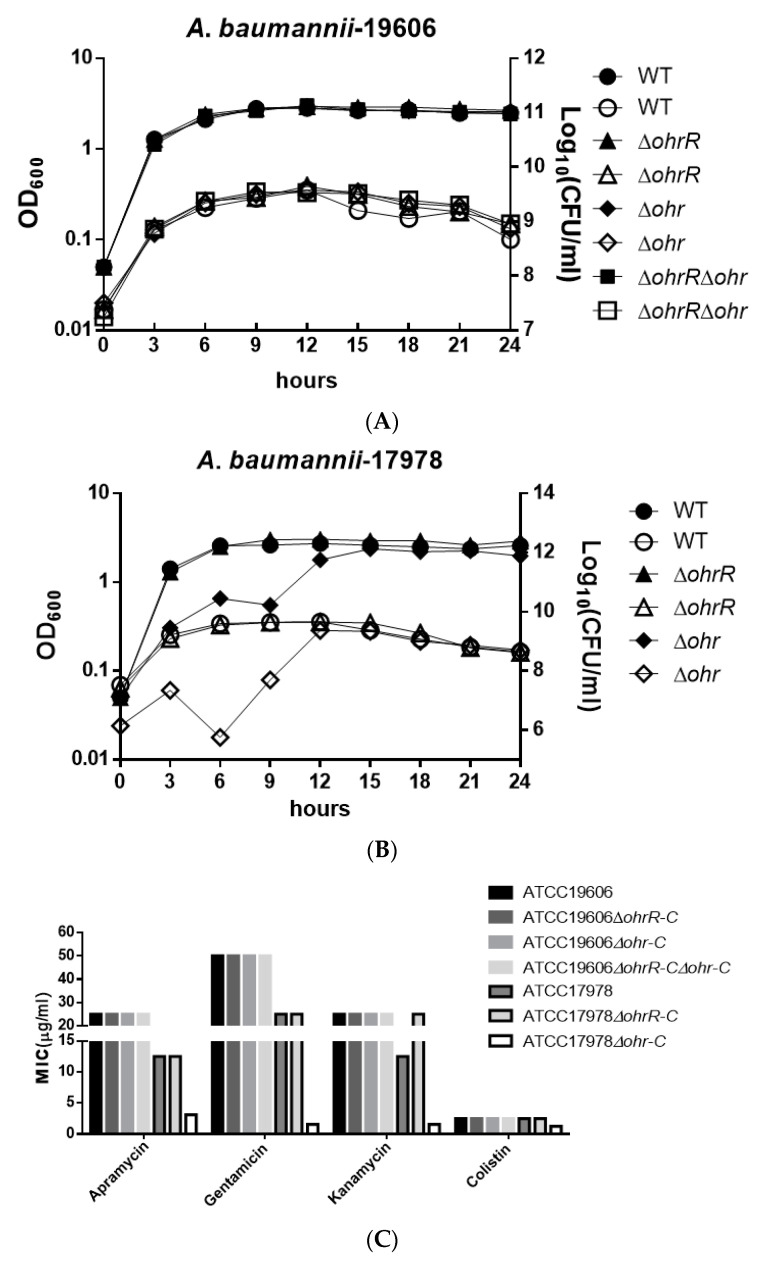
Growth curve of different *A. baumannii* strains in LB medium and minimum inhibition concentration (MIC) of different strains treated by different antibiotics. Growth curve of (**A**) *A. baumannii* ATCC 19606 and (**B**) *A. baumannii* ATCC 17978 wild-type and mutant strains. The *Y*-axis at left represents OD_600_, while the *Y*-axis at right represents viable cell count. Filled symbols represent the optical density and empty symbols indicate the viable cell counts of each strain. These data were obtained from three independent experiments. (**C**) MIC of different strains.

**Figure 4 microorganisms-09-00629-f004:**
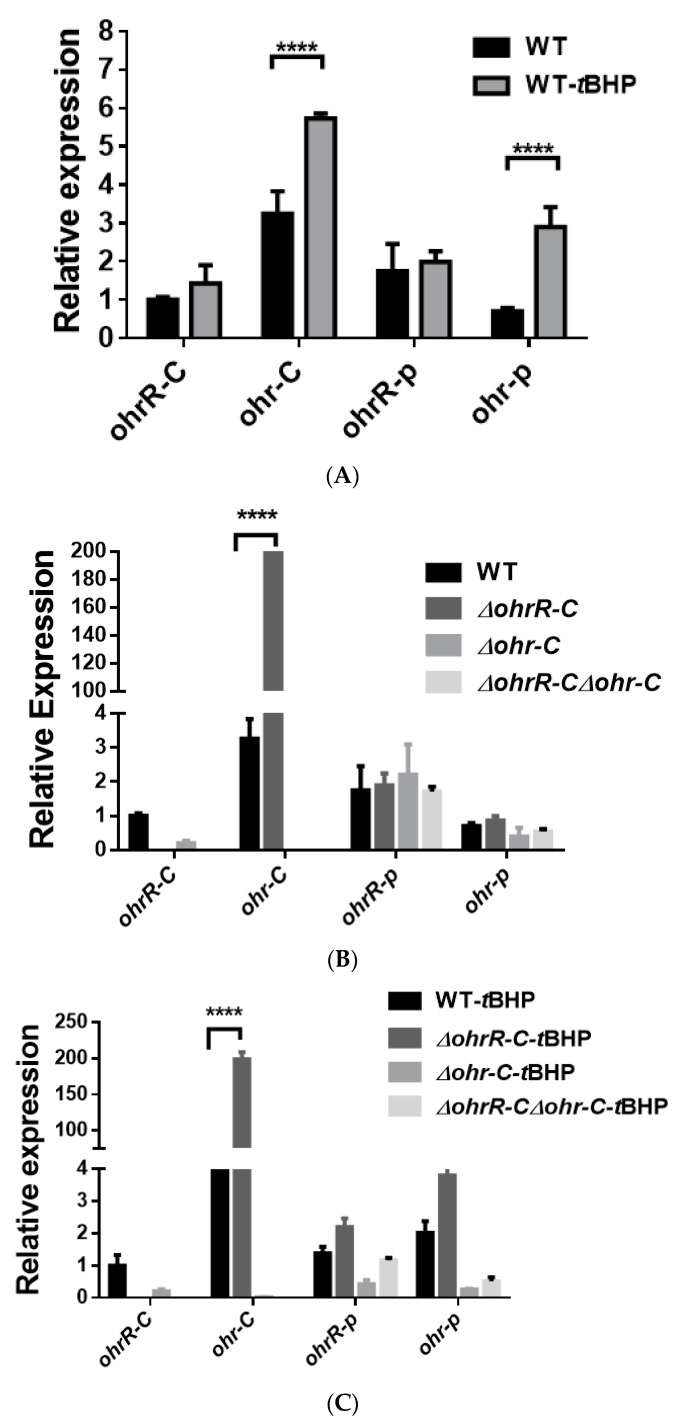
Transcriptional expression of chromosomal *ohrR-C-ohr-C* and pMAC *ohr-p-ohrR-p* genes in different strains of *A. baumannii* ATCC 19606, as quantified by qRT-PCR. (**A**) Relative expression of chromosomal *ohrR-C-ohr-C* and pMAC *ohr-p-ohrR-p* genes in wild-type strains cultured in the presence (WT-*t*BHP) or absence (WT) of 200 μM of *t*BHP for 20 min. Relative expression of chromosomal *ohrR-C-ohr-C* and pMAC *ohr-p-ohrR-p* genes in *ohrR-C* mutant (*∆ohrR-C*), *ohr-C* mutant (*∆ohr-C*), and *ohrR-C-ohr-C* double mutant (*∆ohrR-C∆ohr-C*) strains compared with wild-type strains in the absence (**B**) and presence (**C**) of *t*BHP. These data were obtained from three independent experiments. Each sample was normalized using *gyrase* gene expression as an internal control. The expression of *ohrR-C* gene in wild type was determined as 1 for comparison. Multiple-way ANOVA was used to determine the significance of each phase. **** indicates *p* < 0.001.

**Figure 5 microorganisms-09-00629-f005:**
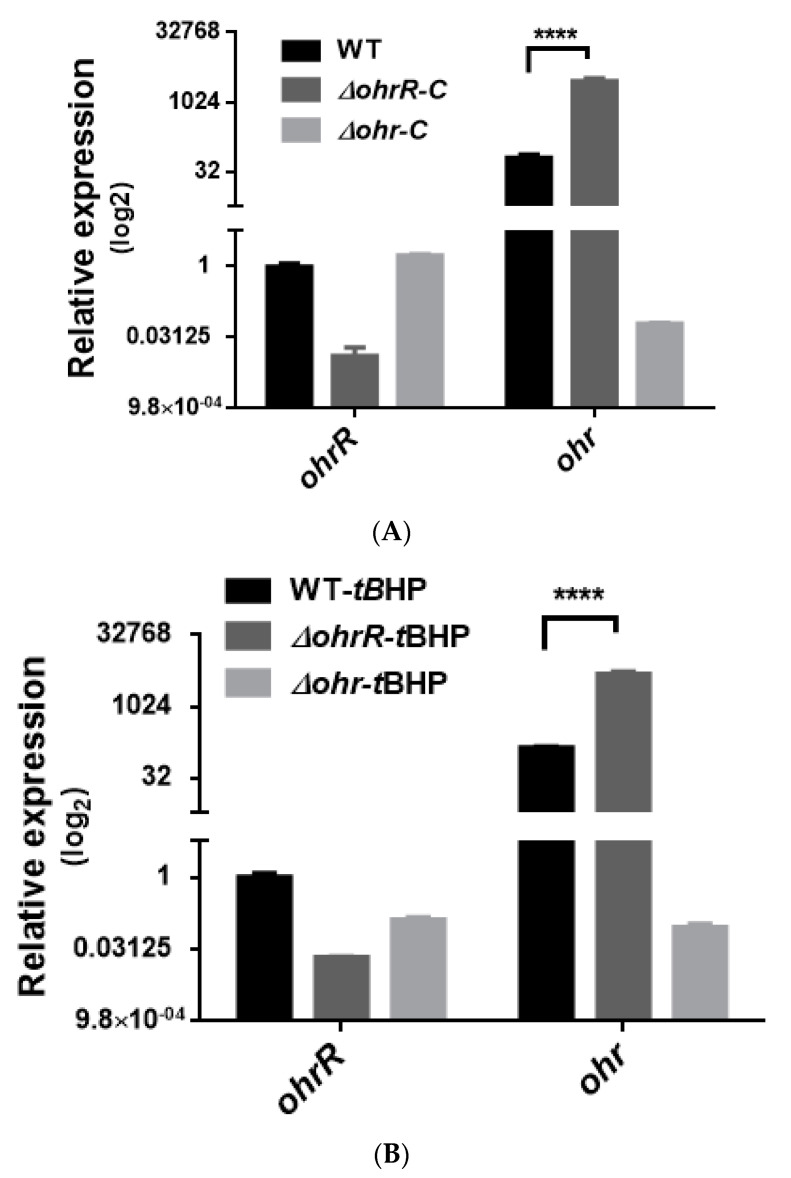
Transcriptional expression of chromosomal *ohrR-C-ohr-C* genes in different strains of *A. baumannii* ATCC 17978, as quantified by qRT-PCR. Relative expression of chromosomal *ohrR-C-ohr-C* genes in *ohrR-C* mutant (*∆ohrR-C*) and *ohr-C* mutant (*∆ohr-C*) strains compared with wild-type strains in the absence (**A**) and presence of 200 μM of *t*BHP for 20 min (**B**). Each sample was normalized using *gyrase* gene expression as an internal control. The expression of *ohrR-C* gene in wild type was determined as 1 for comparison. These data were obtained from three independent experiments. Multiple-way ANOVA was used to determine the significance of each phase. **** indicates *p* < 0.0001.

**Figure 6 microorganisms-09-00629-f006:**
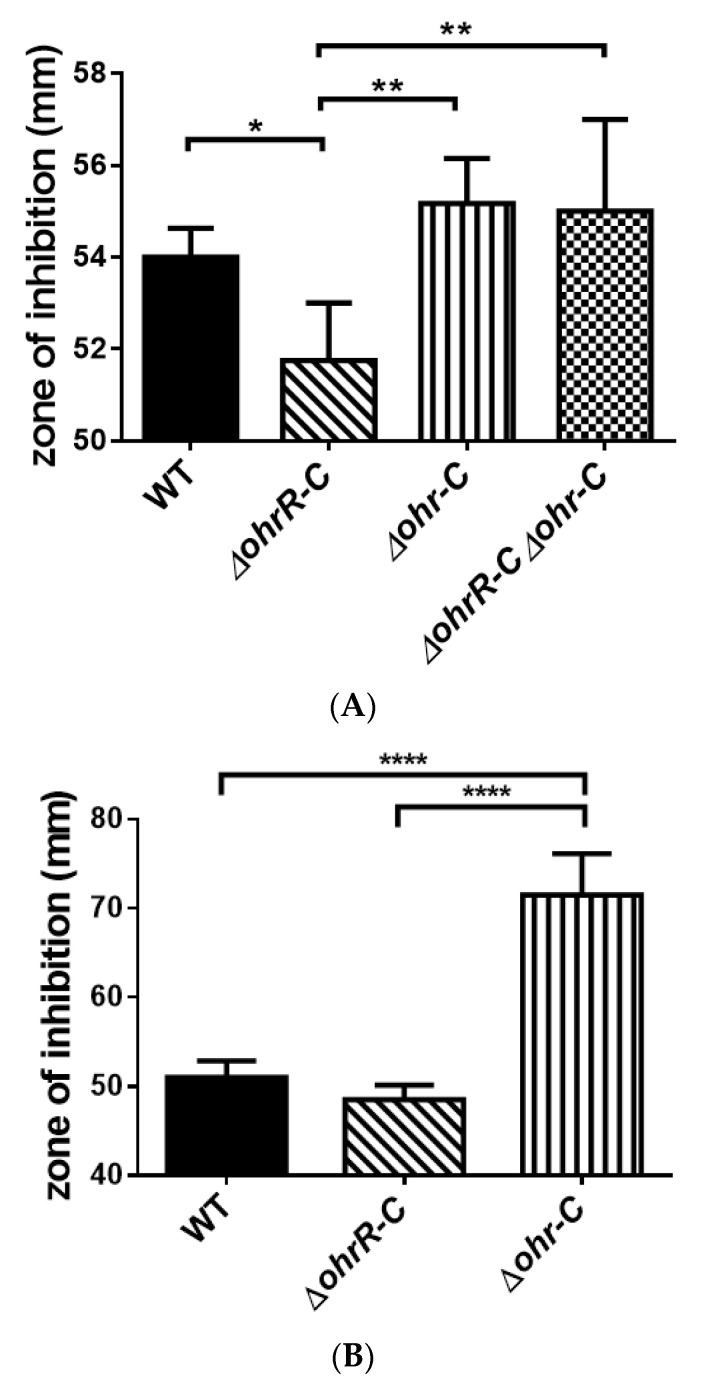
Disk diffusion assay for different strains of *A. baumannii*. (**A**) ATCC 19606 and (**B**) ATCC 17978 disk diffusion assay results. The zone of inhibition was determined after bacteria were treated with 135 μg of *t*BHP for 12 h. These data were obtained from three independent experiments. Multiple-way ANOVA was used to determine the significance of each phase. * indicates *p* < 0.05; ** indicates *p* < 0.01; **** indicates *p* < 0.0001.

**Figure 7 microorganisms-09-00629-f007:**
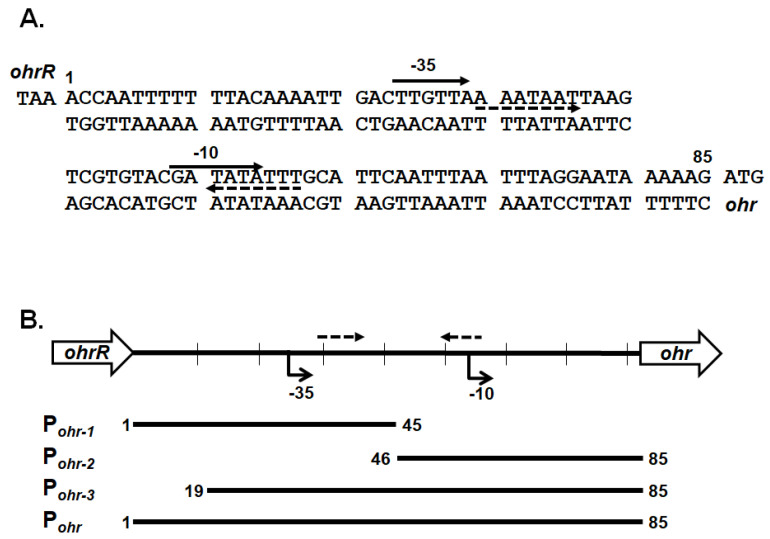
Genetic organization and intergenic sequences for chromosomal *ohrR-C* and *ohr-C* genes. (**A**) Intergenic sequences between the chromosomal *ohrR-C* and *ohr-C* genes. The numbers indicate positions relative to the stop codon of *ohrR*. Arrows with dashed lines indicate the putative inverted repeats. (**B**) Genetic organization and relative position of probes for EMSA.

**Figure 8 microorganisms-09-00629-f008:**
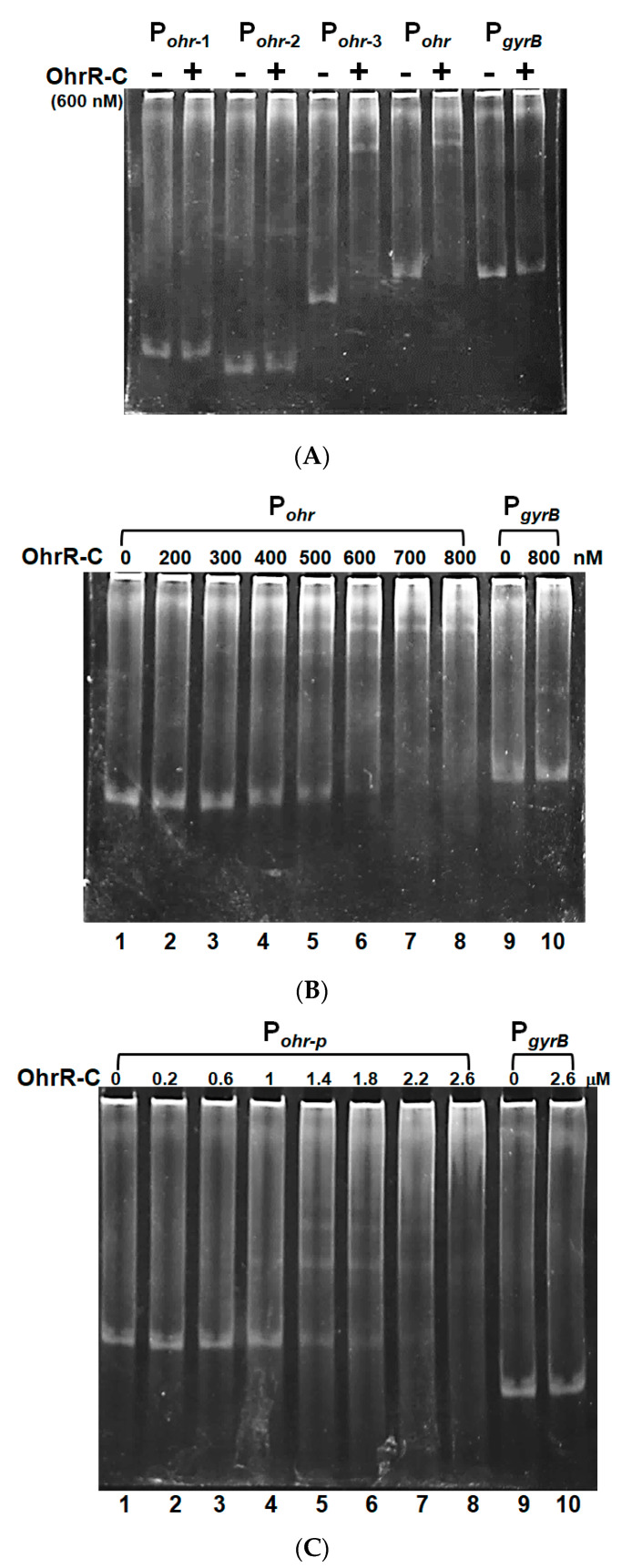
OhrR-C binds specifically to the chromosomal *ohrR-ohr* intergenic region and the *ohr-p* promoter region on pMAC. (**A**) OhrR-C (600 nM) incubated with different probes (50 nM). (**B**) P*_ohr_* incubated with different concentrations of OhrR-C (0–800 nM). P*_gyrB_* is a gyrase gene promoter that was incubated with or without 800 nM of OhrR-C, to serve as a control. (**C**) P*_ohr-p_* interaction with different concentrations of OhrR-C (0–2600 nM). P*_gyrB_* is a gyrase gene promoter that was also incubated with or without 2600 nM of OhrR-C, to serve as a control. Concentrations of OhrR-C are indicated in the top row.

**Figure 9 microorganisms-09-00629-f009:**
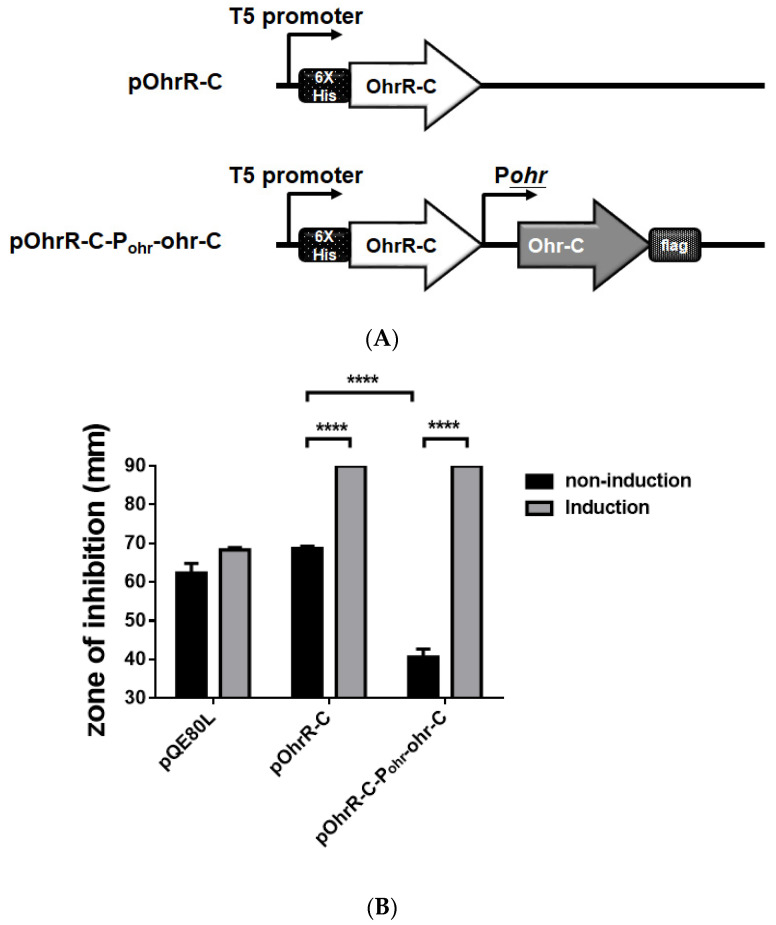
Genetic organization and disk diffusion assay for the gain of function assay in *E. coli.* (**A**) Genetic organization of OhrR-C expressing strains and OhrR-C-Ohr-C expressing strains. (**B**) Disk diffusion assay. Different *E. coli* strains without (black bar) or with (gray bar) 1 mM IPTG induction. Cultures were treated with 135 μg *t*BHP for 12 h. Multiple-way ANOVA was used to determine the significance of each phase. **** indicates *p* < 0.0001.

**Figure 10 microorganisms-09-00629-f010:**
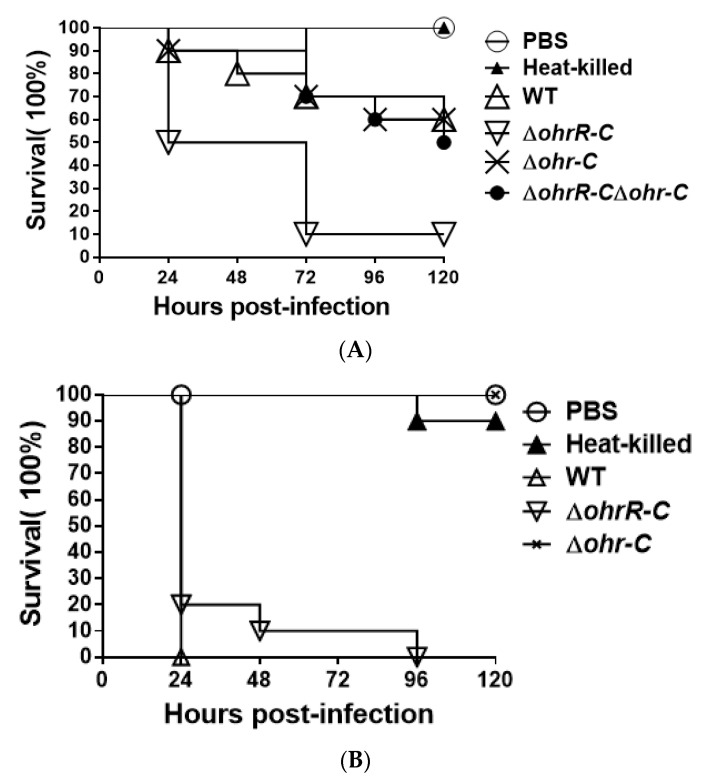
*G. mellonella* survival curve following infection with *A. baumannii* wild-type and mutants. Larvae were infected with 5 × 10^6^ CFU of wild-type or mutant (**A**) ATCC 19606 or (**B**) ATCC 17978 strains. PBS was used as the buffer. Heat-killed indicates wild-type bacteria treated at 100 °C for 10 min. WT, wild type bacteria; *ΔohrR-C*, *ohrR-C* mutant; *Δohr-C*, *ohr-C* mutant; *ΔohrR-CΔohr-C*, *ohrR-C-ohr-C* double mutant. The curve represents a single experiment performed with 10 larvae.

**Table 2 microorganisms-09-00629-t002:** Gene-specific primers used in this study.

Name	Sequences (5′–3′)	Function
*ohrR* MU F	GCTGTGGGTGGATATCAGGA	Construction of *∆ohrR*
*ohrR* MU R	CAGTGACTCGTCTTAAAGAT	Construction of *∆ohrR*
pk18_*ohr*up_F	CGAGCTCGGTACCCGGGCGG GACAACGAATAATTTTG	Construction of *∆ohr*
*ohr*up-*ohr*do-R	CCAATGAAGCAAGTGAAGCA TTATTTAGCTATATTTGTACGGAGC	Construction of *∆ohr*
*ohr*up-*ohr*do-F	GCTCCGTACAAATATAGCTA AATAATGCTTCACTTGCTTCATTGG	Construction of *∆ohr*
*ohr*do-pK18-R	AACGACGGCCAGTGCCATTT ACGTCTTGCTGGTCGTG	Construction of *∆ohr*
pk18_ *ohr*Rup_F	CGAGCTCGGTACCCGGGCAT CTACCCCTTTTGGCAAT	Construction of *∆ohrR∆ohr*
*ohr*Rup- *ohr*do-R	CCAATGAAGCAAGTGAAGCA CCTCAGAAAACTAATGGGTGCT	Construction of *∆ohrR∆ohr*
*ohr* Rup- *ohr* do-F	AGCACCCATTAGTTTTCTGA GGTGCTTCACTTGCTTCATTGG	Construction of *∆ohrR∆ohr*
nptII-F	ATGATTGAACAAGATGGATTGC	Amplification of kanamycin resistant gene
nptII-R	TCAGAAGAACTCGTCAAGAAG	Amplification of kanamycin resistant gene
*gyrase*-PF	AACCTATATTTGCTAGGGAG	Amplification of *gyraeB* promoter region
*gyrase*-PR	TACTAGAGGAATCATAAGCC	Amplification of *gyraeB* promoter region
*ohr*-1 up	ACCAATTTTTTTACAAAATT GACTTGTTAAAATAATTAAGTCGTG	Amplification of *ohr-c* promoter region
*ohr*-1 bo	CACGACTTAATTATTTTAAC AAGTCAATTTTGTAAAAAAATTGGT	Amplification of *ohr-c* promoter region
*ohr*-2 up	TACGATATATTTGCATTCAA TTTAATTTAGGAATAAAAAG	Amplification of *ohr* promoter region
*ohr*-2 bo	CTTTTTATTCCTAAATTAAA TTGAATGCAAATATATCGTA	Amplification of *ohr-c* promoter region
*Ohr*-3_PF	TTGACTTGTTAAAATAATTAAGTCG	Amplification of *ohr-c* promoter region
*ohr*_PF	ACCAATTTTTTTACAAAATT	Amplification of *ohr-c* promoter region
*ohr*_PR	CTTTTTATTCCTAAATTAAA	Amplification of *ohr-C* promoter region
pMAC_ohr_PF	GGCCGATATAAGCTCTATTT	Amplification of *ohr-p* promoter region
pMAC_*ohr*_PR	TGTATATTACCTTGCTTAAT	Amplification of *ohr-p* promoter region
*ohrR* _kpnI_flag-R	GGGGTACCCTTGTCGTCATCGTCTTTG TAGTCTTCAGTCACAATATTAAACG	Construction of pOhrR-C-P_ohr_-ohr-C
*ohrR* F	CGGGATCCATGGACCAAGACTGTCAAA A	Specific primer of chromosomal *ohrR-C*
*ohrR* R	GGGGTACCTTATTTAGCTATATTTGTAC	Specific primer of chromosomal *ohrR-C*
*ohrR* qF	TGGACCAAGACTGTCAAAATC	qRT-PCR primer for *ohrR-C*
*ohrR* qR	TCCCACAACACCAACATCAC	qRT-PCR primer for *ohrR-C*
*ohr* qF-2	AAGCAACAGGTGGCCGTGAT	qRT-PCR and specific primer of chromosomal *ohr-C*
*ohr* qR-2	ACCGACTTCACCTTCAACATACGC	qRT-PCR and specific primer of chromosomal *ohr-C*
ABpMAC_ohrR_F	TGTCCAAGAATCAGCTTTGCT	qRT-PCR and specific primer of *ohrR-p*
ABpMAC_ohrR_R	TTTGTCCAAGATCACCCACA	qRT-PCR and specific primer of *ohrR-p*
ABpMAC_ohr_F	AAAGGTGATGCAACGAATCC	qRT-PCR and specific primer of *ohr-p*
ABpMAC_ohr_R	GTCAAGGCAAATCCACCATT	qRT-PCR and specific primer of *ohr-p*
*gyrB*-F	GGCGGTTTATCTGAGTTTGT	qRT-PCR primer for *gyrase* gene of *A. baumannii*
*gyrB*-R	TTTGTGGAATGTTGTTTGTG	qRT-PCR primer for *gyrase* gene of *A. baumannii*

## Data Availability

All relevant data are within the manuscript and its Appendix A.

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
