# Peer review of "Regulation of tert-Butyl Hydroperoxide Resistance by Chromosomal OhrR in A. baumannii ATCC 19606"

_microorganisms, 2021, doi:10.3390/microorganisms9030629_

Round 1

Reviewer 1 Report

The authors investigated in detail the biological role of OhrR in A. baumannii.

The study is well designed and results are in line with the conclusions.

However, the abbreviations for the chromosomic vs plasmidic location is really confusing. I suggest to change them in Ch-ohrR, Ch-ohr, Pl-ohrR, Pl-ohr. Moreover, it is not clear if Ohr from A. baumannii could be considered the ortologous of B. subtilis OhrA or OhrB. The authors should clarify this point.

In addition, the authors should make some minor English changes in the manuscript and rephrase some sentences.

Author Response

Dear Reviewer 1:

Thanks for your suggestions. Here is the responses to your valuable  suggestions.

1. Thanks for the commend of reviewer. After discussion with all the authors thoroughly, we would like to keep the name of ohr-C, ohrR-C, ohr-p and ohrR-p for C represent the copy in chromosome and p stands for the copy in pMAC plasmid. We also address this in the abstract.

2. After search on sequences and compare the sequences between OhrA, OhrB of  B. subtilis and Ohr-C, Ohr-p of A. baumannii, results showed that Ohr-C demonstrated 38%, 45.3%, amino acid identity with OhrA and OhrB of B. subtilis. However, Ohr-p showed only 38.7%, 38% identity to OhrA and OhrB of B. subtilis. Based on the result, we cannot rule out that Ohr-C might be the ortholog of OhrB from B. subtilis. It still need more studies to figure it out. 

Sincerely, 

Guang-Huey Lin 

Reviewer 2 Report

In this manuscript, Chen et al. describe and characterize the ohrR/ohr genes involved in organic hydroperoxide resistance found in the chromosome of Acinetobacter baumannii as well as in the pMAC plasmid. While this is a well-written, well-rounded paper, I have several major and minor concerns:

  1. In most of their results, the authors fail to specify the number of biological replicates, like in Figure 2, 3, 4, etc. (eg. Does the growth curve represent the number of independent cultures/replicates? How many?). This essential information must be mentioned in the figure legend or materials and methods.

  1. I was unable to identify which Acinetobacter strains/species Figure 2A (numbers 1-20) refer to. Are they all distinct species of Acinetobacter? Which ones? Also, are there genomes sequenced and publicly available from these species to allow a bioinformatic search for ohrR-C, ohr-C, ohr-R-p and ohr-p? That would reinforce the presented data, as PCR results can be affected by mutations in the primer annealing sites and reaction conditions.

  1. Figure 2 and Figure 6 are lacking a complementation strain (ie. the deletion strains with the ohr/ohrR re-inserted) as positive control to determine definitively the ohr genes are responsible for the observed growth, antibiotic sensitivity and tBHP resistance effects.

  1. In the RT-qPCR data (Figure 4A-C), it is unclear what the “relative expression” is relative to. What is the reference strain used for each of these analyses (that is typically normalized to relative expression of 1)? Preferably, the WT without tBHP of each gene should be set as reference.

  1. What happened to the bars for ohrR in Figure 5A? Was it not detected at all or the expression is very much decreased? In the latter situation, setting the y-axis to a log2 should be more suitable for the data presentation. 

  1. Figure 9B indicates that pOhrR-C is constitutively activated in E. coli (“leaky” expression), since even in uninduced cells a lower zone of inhibition is observed. A suggestion to fully assign the sensitivity to ohr-C is cloning the ohr-C gene under the control of a t5 promoter as well and ectopically induce it.

  1. Author state that: “From this gain-of-function experiment, it was observed that OhrR-C was not only able to regulate the expression of the ohr-C gene derived from A. baumannii, but may also suppress endogenous organic peroxide resistance genes in E. coli.” Other than showing ohrR-C overexpression led to higher resistance, there was no evidence to support this claim (eg. no E. coli intrinsic gene expression or promoter binding assays). Many other alternative explanations are possible (eg. Overexpression at non-physiological level per se could have affected the peroxide resistance, or non-specific binding to other unrelated promoters). Authors should either investigate that by performing E. coli gene activation experiments or - since it is not the main point of the manuscript – my suggestion would be to delete/or tone down the sentences that suggest the control of ohrR-C in E. coli intrinsic resistance genes.

Minor:

  1. Authors mention: “Several methods have been applied by our lab (data not shown) and others [20] to evaluate the role of pMAC in organic peroxide resistance by plasmid curing, but these efforts were not successful.” I suggest to be more specific on why it was not successful (curing was not possible, or resistance did not change once plasmid was removed, etc)
  2. Figure 3A and B have been displayed in a way that it is difficult for the reader to distinguish between the separate strain curves for each data point. Data visualization should be improved, perhaps with transparency, with arrows identifying the overlapped curves or even separating the OD from the CFU data into separate panels.
  3. Also, the panel description in the legend of Figure 5 seems to be inverted (panel A is assigned to absence of tBHP, but in the graph it indicates tBHP was added, and vice versa for panel B).

Minor misspellings: 

  • “REFEREMCES” where it should be “REFERENCES”
  • " two conserved cysteines at the catalytic [6].” the word “site” is omitted at the end of the sentence.

Author Response

Dear Reviewer 2: 

Here attach the responses to your valuable suggestion point to point.

Thanks for your kindly suggestions again.

Sincerely, 

Guang-Huey Lin 

Round 2

Reviewer 2 Report

“Since we did not have sufficient budget to acquire Acinetobacter strains with sequencing result available, we isolated bacteria from the environment and identify by 16 rRNA and the other biochemical analysis for those strains. Those strains were applied for tBHP resistant analysis and PCR analysis.”

  • I do understand the budget restriction, however this strain identification procedure is not specified anywhere in the manuscript (eg. Material and methods). These isolates suddenly appear in Figure 2 without any explanation or identification other than numbers. Please describe the identification methods and how close to a species you were able to identify (on line 277, you mention Acinetobacter soli, so I presume you were able to identify at species level some of these isolates). Would you be able to sequence at least some of the 16S rRNA gene amplification products (amplicons) to confirm they belong to Acinetobacter spp.?

“Thanks for the suggestion of reviewers. We already cloned the ohr-C/ohrR-C genes fragment into pWH1266, an Acinetobacter-E.coli shuttle vectors, however we failed to get complementation strains with this complementary plasmid after we have tried to transform this plasmid by electroporation and the other methods for more than one years. That is also the reason why we compare the strain with pMAC (A. baumannii ATCC19606) and without plasmid (A. baumannii ATCC17978) for their tBHP resistance.”

  • This piece of information (the inability of generating a complementary strain) is valuable and should be added to the manuscript’s result and/or discussion – it could suggest ohr copy number/expression levels need to be tightly regulated otherwise it becomes toxic, or an incompatibility of the plasmid used. Although tempting to assume that the differences observed in the comparison of A. baumannii ATCC19606 versus A. baumannii ATCC17978 is due to the presence of ohr in the pMAC plasmid, these two strains are not isogenic. Therefore, many other genetic variabilities could explain the differences observed and authors should be cautious on their interpretation of their results.

“Answers: Figures 4A was compared each of genes with and without tBHP treatment, indicating ohr-C and ohr-p are really induced by tBHP. In Figure 4B, we compared the expression between strains without tBHP treatment. Results revealed that in DohrR-C mutant the expression of ohr-C was elevated indicating that OhrR-C repressed the expression of ohr-C. In Figure 4C, we compared the expression between strains with tBHP treatment.”

  • I understand the comparison the authors are trying to do, but that is not what the relative expression graph is showing. If that is the case, why are your controls not set to relative expression of 1? For example, in Figure 4C, the ohr-C gene in the wild-type is shown having a relative expression of ~2 fold more expression… relative to what? In other words, a 2-fold increase compared to which sample? Normally, in qRT-PCR experiments, one reference/control sample (wild type) is set to 1 and all other strains are compared against it (hence “relative expression” because their gene expression is relative to the control, and not an absolute quantitation). By these graphs, no sample is set as a reference/control. Please revise the qRT-PCR figures to make sure the data presentation is correct.

  • My suggestion for Figure 5 remains (where a bar that is too small to be seen is shown for ohrR gene expression). In general, for gene expression analysis, showing the y-axis (relative expression) transformed in log2 is much more suitable than showing in a linear range. This is a great example where linear range does not allow the reader to visualize the data because of a disproportional effect between the samples (making the bar too small to be seen), while transforming it to log2 would allow the reader to see it.

“Authors mention: “Several methods have been applied by our lab (data not shown) and others [20] to evaluate the role of pMAC in organic peroxide resistance by plasmid curing, but these efforts were not successful.” I suggest to be more specific on why it was not successful (curing was not possible, or resistance did not change once plasmid was removed, etc).

Answers: The curing was not possible since we can find pMAC after curing process”

  • Please add this justification to the manuscript so the readers understand why it fails.

Author Response

Dear Reviewer:

Thanks for your suggestions. All of your valuable suggestions are replied in the attachment file point by point. 

Thanks for your concern and suggestions. 

Guang-Huey Lin
